# “Pulsed Hypoxia” Gradually Reprograms Breast Cancer Fibroblasts into Pro-Tumorigenic Cells via Mesenchymal–Epithelial Transition

**DOI:** 10.3390/ijms24032494

**Published:** 2023-01-27

**Authors:** Anna Nushtaeva, Mikhail Ermakov, Maria Abdurakhmanova, Olga Troitskaya, Tatyana Belovezhets, Mikhail Varlamov, Tatyana Gayner, Vladimir Richter, Olga Koval

**Affiliations:** 1Institute of Chemical Biology and Fundamental Medicine Siberian Branch of the Russian Academy of Sciences, Akad. Lavrentiev Ave. 8, 630090 Novosibirsk, Russia; 2Institute of Molecular and Cellular Biology, Siberian Branch of the Russian Academy of Sciences, Akad. Lavrentiev Ave. 8/2, 630090 Novosibirsk, Russia; 3Department of Natural Sciences, Novosibirsk State University, Pirogova Str. 2, 630090 Novosibirsk, Russia

**Keywords:** hypoxia, cancer associated fibroblasts, pro-tumorigenic cells, c-MYC, OVOL2, mesenchymal to epithelial transition, HIFs, patient-derived cell culture, breast cancer

## Abstract

Hypoxia arises in most growing solid tumors and can lead to pleotropic effects that potentially increase tumor aggressiveness and resistance to therapy through regulation of the expression of genes associated with the epithelial–mesenchymal transition (EMT) and mesenchymal–epithelial transition (MET). The main goal of the current work was to obtain and investigate the intermediate phenotype of tumor cells undergoing the hypoxia-dependent transition from fibroblast to epithelial morphology. Primary breast cancer fibroblasts BrC4f, being cancer-associated fibroblasts, were subjected to one or two rounds of “pulsed hypoxia” (PH). PH induced transformation of fibroblast-shaped cells to semi-epithelial cells. Western blot analysis, fluorescent microscopy and flow cytometry of transformed cells demonstrated the decrease in the mesenchymal markers vimentin and N-cad and an increase in the epithelial marker E-cad. These cells kept mesenchymal markers αSMA and S100A4 and high ALDH activity. Real-time PCR data of the cells after one (BrC4f_Hyp1) and two (BrC4f_Hyp2) rounds of PH showed consistent up-regulation of TWIST1 gene as an early response and ZEB1/2 and SLUG transcriptional activity as a subsequent response. Reversion of BrC4f_Hyp2 cells to normoxia conditions converted them to epithelial-like cells (BrC4e) with decreased expression of EMT genes and up-regulation of MET-related OVOL2 and c-MYC genes. Transplantation of BrC4f and BrC4f_Hyp2 cells into SCID mice showed the acceleration of tumor growth up to 61.6% for BrC4f_Hyp2 cells. To summarize, rounds of PH imitate the MET process of tumorigenesis in which cancer-associated fibroblasts pass through intermediate stages and become more aggressive epithelial-like tumor cells.

## 1. Introduction

Breast cancer (BC) is the leading cause of cancer death among women worldwide, with about one million new cases diagnosed each year [1]. Advanced stages of the disease with distant metastases usually worsen prognosis and outcome [2]. Understanding the molecular and cellular mechanisms underlying metastatic progression is an important step for the development of new antitumor therapeutic strategies. Tumor progression is determined not only by tumor cells but is also a product of the tumor microenvironment (TME), formed by stromal cells, such as fibroblasts and cancer associated fibroblasts (CAFs), immune cells, endothelial cells and extracellular matrix (ECM) [3]. CAFs are highly heterogeneous subpopulations of tumor stroma cells originating from various cellular precursors and usually derived from normal resident tissue fibroblasts, mesenchymal stem cells, adipocyte, endothelial cells and pericytes [4,5]. The CAF cell morphology in vitro is the typical spindle-like shape with decreased cytoplasmic protrusions [6,7], but several studies showed that CAFs are more numerous, appear elongated and possess abundant stress fibers as compared to normal fibroblasts [8,9]. Normal resident tissue fibroblasts in the TME may activate and acquire molecular characteristics of CAFs through the stimulation of different modulators (both molecular and physiological). The various types of modulators described include: transforming growth factor (TGF)-β, hepatocyte growth factor (HGF), platelet-derived growth factor (PDGF), stromal-derived factor-1 (SDF-1) and reactive oxygen species [4]. The transition of fibroblasts, located in close proximity to malignant cells, to the state of activated CAFs involves a loss of CD34 and CD45 antigens and the gain of mesenchymal markers, such as vimentin, α-smooth muscle actin (α-SMA), S100 calcium binding protein A4 (S100A4), fibroblast-activation protein (FAP), platelet-derived growth factor receptor α/β (PDGFRα/β), podoplanin (PDPN) and etc. [10]. By far, one of the greatest challenges in defining CAFs is the lack of a specific biomarker. Standardized functional and molecular definitions of CAFs subtypes also do not yet exist [11], and usually CAFs are differentiated by the expression level of a combination of markers of stromal cells [12]. The primary CAFs should be negative for epithelial (EpCAM), endothelial (CD31) and hematopoietic (CD45) with an elongated spindle-like morphology [13]. Transformation of TME fibroblasts to CAFs is a part of the initial stage of the metastatic cascade, before cell migration and invasion [14].

The epithelial–mesenchymal transition (EMT) and the reverse process, mesenchymal–epithelial transition (MET), are essential processes of transition between the epithelial and mesenchymal cellular phenotypes and defined as a cell’s ability to switch between these two phenotypes, with epithelial cells losing their epithelial characteristics while gaining those of a mesenchymal cell, and vice versa [15]. EMT is thought to be critical for the initial transformation from benign to invasive carcinoma, whereas MET is critical for the later stages of metastasis [16,17]. Subsequent colonization in distant organs requires the reversion of EMT and/or activation of the MET program to establish secondary tumors [18,19]. Several studies showed that an MET-like process can occur in human lung fibroblasts, either from normal or diseased lungs [20], in dermal fibroblasts [21] and for induced pluripotent stem cells [22]. Hypoxia is an essential modulator of the TME by activating hypoxia-inducible factors (HIFs) [23]. HIFs are heterodimers composed of HIF-α (isoforms HIF-1α, HIF-2α, HIF-3α) and HIF-β (also known as aryl hydrocarbon receptor nuclear translocator [ARNT]) subunits that belong to the Per-ARNT-Sim (PAS) family of basic helix–loop–helix (bHLH) transcription factors [24]. HIF-1β is a common binding partner for other members of the family, and it is constitutively expressed. HIF-1α is unique to HIF-1, and its expression is primarily regulated by oxygen tension [25]. The constitutive HIF-1β/ARNT subunit and the highly oxygen-sensitive HIF-1α subunit constitute the HIF1 protein, while intracellular HIF-1α levels determine the activity of HIF-1 [26]. Tumor hypoxia regulates gene expression, apoptosis and autophagy in the tumor and the cells from the TME [27], stimulating EMT and MET, enhanced angiogenesis and vasculogenesis [28], and cellular metabolism [29]. Most notably, MET has been shown for fibroblasts in ontogenesis as a fundamental process. During the development of an organism, fibroblasts change their mesenchymal phenotype to an epithelial one [30]. MET involves a complex functional phenotypic change from a typical mesenchymal non-polarized cell to a polarized epithelial-like cells. 

The expression of EMT- and MET-related genes helps to adapt cells to hypoxic conditions [31]. The mechanism of EMT has been extensively studied over decades, and several transcription factors (TFs) were identified as master-regulators that belong to the c-MYC, SNAIL/SLUG, ZEB and TWIST families [32]. While many upstream regulators of EMT have been identified, very little is known about the regulation of MET [16,33]. MET has recently drawn attention as an attractive molecular pathway for targeted anticancer therapy [34]. Most of the current MET studies are based on EMT-TF’s activity and in vitro effects on animal models [33,35]. Several studies showed that TFs of the OVOL family regulate MET in several cancers, such as prostate, breast, lung, cutaneous squamous and cutaneous squamous carcinomas [36,37,38]. In our thorough analysis of worldwide researches, we did not find any studies on the topic of spontaneous MET events in breast cancer cell lines. Thus, the spontaneous MET was described for cell line PMC42 mesenchymal breast carcinoma, which can be transformed into epithelial cell line PMC42-LA [39,40]. For other types of cancerous cells, Becerril et al. observed that some human lung fibroblasts under standard conditions or with conditioned media were transformed into a lineage of epithelial-like cells by an MET-like process [20].

In the current study, we used the fibroblast cell line BrC4f, which was isolated from breast adenocarcinoma tissue and propagated as personal cell culture. Using a ”pulsed hypoxia” method, the intermediate cell lines BrC4f_Hyp1 and BrC4f_Hyp2 with transitional “fibroblast-to-epithelial” phenotypes were obtained. These cells imitate tumor cells undergoing the hypoxia-dependent transition from fibroblast to epithelial morphology [41]. The following research questions were set by the study: (i) whether the BrC4f fibroblast cell line belongs to CAFs, (ii) whether it is possible to obtain an intermediate phenotype as a model to study the molecular features of such an intermediate form, (iii) how the transformation into epithelial-like tumor cells changes EMT- or MET-associated cellular markers and (iv) are the intermediate phenotypes BrC4f_Hyp1 and BrC4f_Hyp2 more aggressive in vivo then parental BrC4f.

## 2. Results

### 2.1. Hypoxia-Induced Changes in Cell Morphology of a Fibroblast Culture

In a previous study, we found that specific culturing conditions (“pulsed hypoxia” (PH) and conditioned media) progressively transformed cells from mesenchymal to epithelial phenotypes. This transformation was accompanied by the changes in respective molecular markers, proving that PH activated mesenchymal-to-epithelial transition [41]. “Pulsed hypoxia” is based on the repetitive switching of the cultivating condition from normoxia to hypoxia. The ratio of oxygen portion in the medium during “pulsed hypoxia” changes by 7%.

Here, we used the fibroblast cell line BrC4f to subject to the “pulsed hypoxia” method for transformation of the cells’ morphology from fibroblast to epithelial. These cells were prepared from breast adenocarcinoma tissue, as was described previously in Ref. [41]. One round of hypoxia (48 h) resulted in BrC4f_Hyp1 cells, and two rounds of hypoxia (96 h) resulted in BrC4f_Hyp2 cells. Figure 1 demonstrates the changes in BrC4f morphology along rounds of ”pulsed hypoxia”. The transformation of fibroblast cells to epithelial cells can be clearly seen. The parental BrC4f cells displayed a typical fibroblast-like phenotype, the BrC4f_Hyp1 culture contained the islets of epithelial-like cells, and in the BrC4f_Hyp2 culture, the epithelial-like phenotype prevailed over the population of fibroblast-like cells. Further cultivation of BrC4f_Hyp2 under normoxia conditions led to full replacement of fibroblasts to epithelial cells, and these cells were named BrC4e. Under standard cultivation conditions, BrC4e cells were characterized by cell–cell contacts, focal accumulation and a cobblestone shape (Figure 1a). 

The alteration of HIF-1β/ARNT and HIF-1α proteins in parental and PH-generated cell lines was analyzed by Western blots analysis of these cell cultures (Figure 1b). Surprisingly, the signal, with corresponding HIF-1α proteins, showed that fibroblast-like BrC4f cells were expressed in normoxia. The HIF-1α protein level increased after two rounds of PH and reduced in BrC4e cells compared to BrC4f cells. No changes were observed in the levels of HIF-1β/ARNT along rounds of PH and a significant increase in BrC4e cells (Figure 1b). 

### 2.2. PH-Induced Transformation of Fibroblast Follows a Change in Mesenchymal Markers Including Cytoskeletal Markers and an Increase in E-Cadherin 

To identify the molecular changes in BrC4f_Hyp1, BrC4f_Hyp2 and BrC4e compared to parental BrC4f cells, we analyzed the mesenchymal markers vimentin and N-cadherin and the epithelial marker E-cadherin by western blotting (Figure 2a) and flow cytometry (Figure 2b). One round of “pulsed hypoxia” for BrC4f decreased vimentin’s and simultaneously increased N- and E-cadherin’s expression. This tendency stayed the same for BrC4f_Hyp2 cells passed through the second round of “pulsed hypoxia”. Stabilization of the cultivating condition to normoxia resulted in BrC4e, which showed the divergence in N- and E-cadherin expression: these cells were semi-negative for N-cadherin and high positive for E-cadherin. Moreover, as in classical epithelial cells, these cells were vimentin-negative (Figure 2 and Appendix A). Thus, the phenotypes of BrC4f_Hyp1 and BrC4f_Hyp2 cells can be classified as intermediate mesenchymal/epithelial molecular phenotypes relative to the initial fibroblasts BrC4f and the resulting epithelial-like cells BrC4e.

Since fibroblasts and epithelial cells differ in the proteins that form the cytoskeleton, and such proteins are considered as markers of mesenchymal or epithelial phenotype, it was important to identify such proteins in the examined cell lines. Alpha-smooth muscle actin (αSMA) and S100 calcium binding protein A4 (S100A4) were analyzed in Brc4f CAFs and BrC4e epithelial-like cells. Breast carcinoma MDA-MB-231 mesenchymal-like cells and epithelial MCF7 cells were used as internal positive and negative controls, respectively. Surprisingly, immunofluorescence staining showed that fibroblast-like BrC4f and epithelial BrC4e cells were positive for both αSMA and S100A4, while MCF7 cells were αSMA-negative. Thus, BrC4e retained mesenchymal cytoskeleton markers after MET, although they lost their spindle shape (Figure 3). 

### 2.3. Identification of the Molecular Markers of Cancer-Associated Fibroblasts in BrC4f Cells 

The ability to undergo mesenchymal–epithelial transition can be characteristic of CAFs. Therefore, we made an additional search for CAF markers in BrC4f cells. Cells were immunochemically stained for CD31 [13], CD34 [42], CD44 [43], CD45 [10], CD73 [44], CD90 [45], CD105 [46] and EpCAM [13] and analyzed by flow cytometry. As a negative control, the cells were stained with anti CD34 and CD45 (hematopoietic cells markers), CD31 (endothelial) and EpCAM (epithelial) markers (Figure 4a). Phenotype analysis of BrC4f cells showed high expression of CD90 and CD73 antigens and a lack of the CD105 antigen (Figure 4b). Taking into account the high level of vimentin and N-cadherin in these cells, the data obtained let us classify BrC4f cells as mesenchymal stem cell-like CAFs [43]. 

### 2.4. Alteration of Gene Expression Profile during Epithelial–Mesenchymal Transition and Mesenchymal-Epithelial Transition

Adaptation to hypoxic conditions is regulated by many factors, including the expression of genes associated with epithelial–mesenchymal transition (EMT), for example, SLUG, ZEB1/2 TWIST1 and by gene-regulators of the reverse process, mesenchymal–epithelial transition (MET), for example, OVOL1/2 [31]. Proto-oncogene c-MYC also controls the expression of numerous genes which are essential in the regulation of EMT [47]. We explored the alteration in the expression of typical gene-regulators of EMT and MET in BrC4f cells, in cells with intermediate phenotypes (BrC4f_Hyp1 and BrC4f_Hyp2) and in the BrC4e cells. The ER-positive breast adenocarcinoma cell line MCF-7 was used as a positive control and the epidermoid carcinoma cell line A431 for EMT and MET genes, respectively. Relative mRNA levels of c-MYC, SLUG, ZEB1/2 TWIST1 and OVOL1/2 genes were analyzed by Real Time PCR with specific primers (Figure 5b–g). One round of “pulsed hypoxia” stimulated up-regulation of TWIST1 gene in BrC4f_Hyp1 cells compared to BrC4f cells (Figure 5c). The second round of PH resulted in the increase in ZEB1/2 and SLUG gene expression (Figure 5b,d,e). No differences were detected in OVOL1 mRNA expression between BrC4f cells and their PH-generated offspring cells. 

The alteration of SNAIL and SLUG proteins in parental and PH-generated cell lines was analyzed by immunoblotting of these cell cultures. We used antibodies which simultaneously bind SNAIL and SLUG antigens. The signal, corresponding SNAIL + SLUG proteins, increased up to three times in epithelial BrC4e cells in comparison with parental BrC4f cells. (Figure 5a). Thus, we observed significant changes in the mRNA levels of MET regulatory genes during the transformation of the phenotype from fibroblast to epithelial.

### 2.5. Mesenchymal–Epithelial Transition Results in the Modifications of Cellular Metabolism 

Hypoxic environment is also an important factor for the metabolic rewiring in breast cancers. Enzymes that involved in oxidative phosphorylation/glycolysis and enzymes involved in the recycling of nutrients under extremal conditions in the cell can respond to hypoxia. Cathepsin D (CatD) is a lysosomal protease which hydrolyses proteins during autophagy aimed at tumor cell survival under stress conditions. Glyceraldehyde-3-phosphate dehydrogenase (GAPDH) catalyzes transformation of glucose in glycolysis cycle [48,49]. The aldehyde dehydrogenase 1 (ALDH1) enzyme, which is involved in the synthesis of retinoic acid, has been identified as functional stem cell marker in a variety of cancers [50]. Here, we analyzed the changes in GAPDH, ALDH and CatD during the PH-activated gradual transformation of BrC4f cells. The GAPDH level in BrC4f and in derived cells was analyzed by western blot. We found the gradual decrease in GAPDH in the row from CAF-like parental BrC4f cells to epithelial BrC4e cells (Figure 6a). ALDH activity in the cells was analyzed by flow cytometry using the Aldefluor reagent (Figure 6b). Our studies have demonstrated that ALDH activity was reduced in cells that were exposed to hypoxic conditions (BrC4f_Hyp1 and BrC4f_Hyp2) and raised in epithelial BrC4e cells, cultivated in normoxia after rounds of hypoxia.

Cathepsin D activity was estimated in cell lysates using a fluorescent substrate-basing kit by fluorometric technique. We found that the maximum activity of CatD in BrC4f_Hyp2 was up to seven times greater than in BrC4f cells. The lowest level of CatD activity was in epithelial BrC4e cells (Figure 6c). Thus, we can assume that the transformation of BrC4f cells into BrC4e cells is accompanied by the activity of proteolytic enzymes, in particular, CatD. 

### 2.6. In Vivo Tumorigenic Properties of Established Cell Lines 

The aggressive phenotype of cancer cells is usually manifested by a rapid growth in the tumor node and high invasiveness. We tried to assess the tumorigenicity of the cancer-associated fibroblasts BrC4f and BrC4f_Hyp2 with an intermediate phenotype when they were subcutaneously transplanted onto immunodeficient NOD/SCID mice. Tumor growth was monitored twice a week. There was no animal mortality observed during the experiment. In a previous study, we showed that BrC4e cells do not contribute to tumor formation [41].

Data obtained demonstrate that the tumors that were initiated by the BrC4f_Hyp2 cells grown at a greater rate compared to those that were initiated by BrC4f CAFs. Tumor nodules began to be palpated on the 15th day after transplantation of BrC4f_Hyp2 cells and on day 24 for BrC4f cell transplants (Figure 7a). Tumor growth rate (TGR) was calculated to estimate tumor progression and was ultimately equal to 61.6% (Figure 7b). On day 47 after the cell transplantation, all of the tumors were removed and fixed for the subsequent histologic analysis. Histologic analysis of tumors revealed that the shape of tumors resembled breast tissue where lobules of the mammary gland were visible. Tumors formed by the parental BrC4f fibroblasts contained few collagen fibers and epithelial islets (Figure 7b, arrow) and were reach areas of fibroblasts. In contrast, tumors formed by BrC4f_Hyp2 cells represented carcinoma-like morphology with central necrosis zona. Developed tumors demonstrated cellular atypism manifested by the changes in the nucleus to cytoplasm ratio, hyperchromicity and multiple mitosis. These alterations are typical for fast-growing tumors. Thus, the tumors formed by BrC4f and by BrC4f_Hyp2 cells not only grew at different rates, but also had different structures.

The c-MYC level was estimated immunohistochemically in tumor sections. High c-MYC signal was observed in BrC4f_Hyp2 tumor sections. Quantification of the c-MYC oncoprotein in tumors revealed a two-fold increase in PH-generated BrC4f_Hyp2 cells in comparison with parental BrC4 cells (Figure 7d). Data on the increased TGR of tumors formed by BrC4f_Hyp2 cells are in good agreement with RT-PCR data at the level of c-MYC in BrC4e cells cultures.

## 3. Discussion

The term “hypoxia” is used to describe the oxygen scarcity or low-oxygen environment arising in different biological processes, including embryogenesis and tumor growth [51,52]. Hypoxia may promote morphological transformations in various cell types through regulation of epithelial–mesenchymal transition (EMT) or mesenchymal–epithelial transition (MET) by the pathway of hypoxia inducible genes. In stromal fibroblasts, such transformations result in cancer-associated fibroblasts (CAFs) with changes in fundamental functions and structure [29,53,54]. In order to mimic the variation in oxygen level in a tumor environment, there is a need for elaboration of physiologically relevant in vitro models representing the “floating” transition from hypoxia to normoxia [55].

In a previous study we have demonstrated that PH induces MET transformation in primary fibroblast-shaped breast cancer cells BrC4f into BrC4e cells with an epithelial-like phenotype. Genetic analysis and G-banding of BrC4e cells found near triploid karyotype and multiple chromosomal rearrangements, and such rearrangements can also contribute to tumor heterogeneity and clonal evolution as mechanisms for metastasis and drug resistance (Appendix A) [56]. The key role of hypoxia in the transition of BrC4f-to-BrC4e cells was confirmed by the fact that the spontaneous MET was not found in BrC4f cells. These results stimulated us to search the cells with intermediate MET phenotypes. Indeed, in this study, by applying a limited number of rounds of pulsed hypoxia to BrC4f cells, we obtained cells that visually correspond to the intermediate phenotype–BrC4f_Hyp1cells and BrC4f_Hyp2 cells (Figure 1). Further, it was important to determine in which order the activation/deactivation of MET-associated molecular markers occurs. We demonstrated that vimentin-rich BrC4f fibroblasts gradually reduced vimentin in BrC4f_Hyp1 cells and BrC4f_Hyp2 cells and completely lose it in the epithelial BrC4e cells (Figure 2). Such molecular transformations are in good correlation with the MET-like process and, apparently, help to stabilize cells under cultivating conditions from hypoxia to normoxia [20]. Next, we tried to detect N- to E-cadherin switching in cell lines. Analysis of E-cadherin in the original BrC4f cells and in their PH-generated offspring also showed a linear increase in positivity of this epithelial marker from BrC4f fibroblasts to BrC4e. However, changes in N-cadherin level in the original and transformed cells were non-linear in the series of cellular phenotypes from BrC4f to BrC4e. Surprisingly, in cells with intermediate phenotypes, along with an increase in E-cadherin, N-cadherin also increased. Substantial reduction in N-cadherin was observed only in epithelial BrC4e cells. It can be assumed that BrC4f_Hyp1 cells and BrC4f_Hyp2 cells retain the ability to reverse transition to fibroblasts depending on the environment, whereas the phenotype of BrC4e is stably epithelial (Figure 2). Hollestelle et al. have demonstrated that in human breast cancer, loss of E-cadherin is not causal for EMT and is not even a necessity [57]. By analogy, we can assume that loss of N-cadherin is not necessary for the early stages of MET.

The stroma of breast cancer tissues has been shown to be complicated in cellular content and is where cancer-associated fibroblasts represent the most abundant component [58]. We assumed that BrC4f cells, with their potency and plasticity to MET, may belong to the CAF-type [5]. Analysis of CAF-specific markers [11] showed that the molecular phenotype of BrC4f cells was CD73^high^/CD90^high^/CD44^high^/Vim^+^/α-SMA^+^/N-Cad^+^/S100A4^+^/CD105^neg^/CD34^neg^/CD45^neg^/CD31^neg^/EpCAM^neg^ and corresponded to CAF’s and stem cell’s potency (Figure 2, Figure 3 and Figure 4). Genetic data of various melanoma, breast, prostate, colorectal and ovarian cancer cells with the CAF phenotype are controversial: some authors reported that CAFs are diploid and do not acquire genetic changes [59,60], while others showed chromosomal rearrangements and gene copy number alterations [61]. CAFs from solid carcinomas directly modulate tumor growth by secreting factors with oncogenic and mitogenic functions. To date, the relationships between metastasis of breast cancers and the expression of α-SMA and S100A4 by CAFs have been described. CAFs with concurrent expression of α-SMA and S100A4 in breast cancers were associated with lymph node metastases [62]. In breast carcinoma mouse models, in vivo experiments demonstrated the role of S100A4 expressed by stromal fibroblasts to stimulate metastasis formation [63]. We assume that α-SMA and S100A4 retention in the BrC4e cells after the MET transition promotes the aggressive metastatic phenotype of these cells (Figure 4). Additionally, the fibrillar collagen receptor DDR2 (discoidin domain receptor 2) mRNA was up-regulated in BrC4f_Hyp1 cells and BrC4f_Hyp2 cells in comparison with BrC4f and BrC4e cells (Appendix A). It is known that the increase in the endogenous DDR2 mRNA in CAFs switches them into an active state and regulates “fibroblast-to-epithelial” transition during breast cancer development and progression [64]. Thus, PH stimulates the activation of BrC4f CAFs with up-regulation of DDR2 mRNA in the “intermediate” cell lines (Appendix A). 

HIF-1α and HIF-1β are main regulators of the cellular responses to oxygenation and are often referred to as “master regulators of hypoxic responses” [65]. While HIF-1 alpha’s stability is dependent upon oxygen conditions, HIF-1 beta is stable in both normoxia and hypoxia [66]. We found HIF-1α in parental BrC4f cells and PH induced an increase in HIF-1α level in BrC4f_Hyp2 cells. The return to normoxia promoted a subsequent decrease in HIF-1α and increase in HIF-1β in BrC4e epithelial cells (Figure 1b). Zhang et al. demonstrated that HIF-1α is essential for the activation and tumor-promotion function of CAFs in lung cancer [67]. Other studies suggest that HIF-1α induces the metabolic reprogramming of CAFs and increases glycolysis, thereby promoting tumor growth in breast cancer [68,69]. HIF can also indirectly regulate cellular processes such as proliferation and differentiation through interactions with other signaling proteins such as c-MYC and Notch [70]. We assume that HIF-1α level in BrC4f cells can relate to the CAF phenotype.

According to the EMT/MET-associated gene expression in established cell lines, PH activated SLUG, TWIST1, ZEB1 and ZEB2 genes in intermediate BrC4f_Hyp1 cells and BrC4f_Hyp2 cells and more substantial mRNA increase were found for ZEB1/2 genes (Figure 5). Reversion to normoxia conditions with a visible metamorphosis of BrC4f_Hyp2 cells to BrC4e strongly downregulates SLUG, TWIST1 and ZEB1/2 genes. Thus, cells with an intermediate phenotype allowed us to state that during MET, there is no quick suppression of SLUG, TWIST1 and ZEB1/2 genes, as one would assume based only on the initial fibroblasts BrC4f and the final BrC4e epithelial cells. We suggest that activation of the expression of these genes in intermediate BrC4f_Hyp1 cells and BrC4f_Hyp2 cells regulates the MET observed.

Lundgren and colleagues found that hypoxia induces SNAIL expression and activates the migratory potential of mesenchymal-like cell lines [71]. It can be assumed that SLUG expression is needed to stabilize the fibroblast phenotype in BrC4f_Hyp2 cells, whereas in BrC4e cells, the decrease in SLUG mRNA leads to the abolition of E-cadherin transcriptional repression and helps to achieve a low-motility epithelial state. 

In contrast, OVOL2 and c-Myc genes were suppressed in fibroblasts and in the cells with intermediate phenotypes, and their up-regulation was observed only in the epithelial-like BrC4e cells (Figure 4f,g). It is most likely that the expression of these genes is predominantly needed to “fix” the epithelial phenotype of rapidly dividing tumor cells. This is a reasonable assumption, given that OVO-like genes are master regulators of the fate of epithelial cells in breast cancer. Garcia et al. have also shown that Ovol1 and Ovol2 modify the transcriptome of mesenchymal breast cancers cells by downregulation of the mesenchymal markers [72]. Moreover, Wu et al. have shown that OVOL2 suppressed stem cells population (CD44+/CD24−) in mesenchymal MDA-MB-231 cells [73]. We have also found a high number of CD44+/CD24− cells in BrC4f CAFs and in cells with transitional phenotypes (Appendix A) that may correlate with their potential to migrate to distant metastatic locuses. Thus, we showed that transformation of BrC4f CAFs into epithelial-like BrC4e cells during rounds of pulsed hypoxia demonstrated a MET-like process with intermediate molecular cell phenotypes during metastasis, thereby producing a suitable phenotype for the colonization of a metastatic site. 

In solid tumors, the metabolic transformation is the result of complex interactions between a generally hypoxic TME and multiple oncogenic mutations, which drive the alterations in cellular metabolism which occur in transformed cells [74]. In our study, PH conditions consistently suppressed GAPDH content in the range of cellular phenotypes from BrC4f CAFs to epithelial-like BrC4e cells. Results on the hypoxic regulation of GAPDH are contradictory and seem to be type-specific for various tumor cells. Increased expression of GAPDH is observed in several tumors such as prostate, breast, lung and cervical carcinomas, however, there is no alteration in expression in tumors such ashepatoma, colon cancer, lung adenocarcinoma and glioblastoma. Therefore, it is necessary to interpret the data on GAPDH in the context of other metabolic enzymes [75]. For example, ALDH oxidizes intracellular aldehydes and decreases their potential toxicity, so it plays an important role in oxidative metabolism [76]. Moreover, aldehyde dehydrogenase (ALDH) activity is recognized as a biomarker of breast cancer stem cells with a CD44+/CD24− phenotype [77] (Appendix A). However, we have shown that hypoxic conditions suppress ALDH activity in fibroblast and intermediate cells, while the return to normoxia noticeably amplifies ALDH activity in epithelial-like BrC4e cells (Figure 5b). 

Another option to overcome oxygen starvation is by way of active catabolism. Lysosomal protease cathepsin D is a marker of poor prognosis in breast cancer [78]. We found that the highest activity of CatD was in BrC4f_Hyp2 cells (Figure 5c). Reversion of BrC4f_Hyp2 to normoxia conditions converted them to epithelial-like BrC4e cells with reduced CatD activity. We assume that cells such as BrC4f_Hyp2 may contribute significantly to the poor prognosis in breast cancers. Our finding that CatD activity was substantially higher in PH-stimulated BrC4f_Hyp2 cells compared to parental BrC4f and epithelial-like BrC4e cells confirms the potential of BrC4f_Hyp2 cells to destabilize the extracellular matrix and migrate. It can be assumed that in a solid tumor it is cells such as BrC4f_Hyp2 that become capable of leaving their main tumor node to move to a new settlement niche and resemble a highly malignant phenotype. Thus, the return to normoxia stimulates the transition to the epithelial state, through, among other means, suppression of GAPDH and activation of ALDH. 

To confirm our hypothesis that BrC4f_Hyp2 acquired a more aggressive phenotype than parental cells, in vivo tumorigenicity studies of BrC4f_Hyp2 and BrC4f were performed. As expected, palpable tumors appeared much earlier in the case of BrC4f_Hyp2 cells, and the acceleration of tumor growth (TGR) was equal to 61.6%. Thus, animal experiments confirmed that BrC4f_Hyp2 cells demonstrated a more aggressive phenotype than the initial BrC4f cells. Moreover, we observed a significant difference, not only in the growth rate of these two tumors, but also in their histological structure. BrC4f MSC-like CAFs formed a fibrotic focus with islets of epithelial cells. This was expected for cells with CAF characteristics and related them to the high malignancy of a tumor [79]. Li and colleagues demonstrated that the high fibrosis was closely associated with the strong invasiveness and the high malignancy of breast tumors [80]. We assume that the fibrotic focus was formed by clones of the BrC4f, and the epithelial islets were formed by the cells, which, in vivo, undergo MET-like transition. Rapid-growing BrC4f_Hyp2 tumors had no fibrotic focuses and consisted of epithelial-like cells. It is likely that the MET transition in BrC4f_Hyp2 cells was spontaneously completed in the mouse organism under stimulation by endogenous factors. The structures of tumors changed from the invasive BrC4f with an illegible histological type to a homogeneous carcinoid-like morphology of BrC4f_Hyp2 with surrounding lymphoplasmacytic inflammatory infiltrate (Figure 7b) [81]. We found that c-MYC levels were significantly higher in PH-stimulated BrC4f_Hyp2 cells compared to parental BrC4f cells (Figure 7d). Shi and colleagues found that c-MYC expression in fibroblasts can induce epithelial-like morphological changes via MET [82]. Several studies showed that tumors mediated by c-MYC overexpression have a greater propensity for progression and an aggressive phenotype [83,84].

We cannot say with certainty that the BrC4f cells, as a CAF or other breast cancer cell with mesenchymal-like phenotype, have high stem cells potency. By far, one of the greatest challenges in defining CAFs is the lack of a specific biomarker. Expressions of CD90, CD73, vimentin and N-cadherin also reported in MDA-MB-231 cell or other breast cancer cells. EMT/MET programs have been implicated as potential drivers of breast cancer invasion and metastasis, in part by eliciting dramatic cytoskeleton remodeling from characteristic of epithelial cells to the characteristic of mesenchymal cells [85]. MDA-MB-231 cells are epithelial breast cancer cell lines with a mesenchymal-like phenotype derived from metastatic cancers [86]. Wang et al., 2015 showed that that cells of the CD105+/CD90+ subpopulation in MDA-MB-231 accounted for 0.99% by flow analysis [87]. BrC4f cells are positive for CD90 and CD73 for 100% and CD105-negative with strongly spindle-like morphology. It can be suggested that the expression of CD90, CD73, vimentin and N-cadherin in MDA-MB-231 or other breast cancer cells may be related to the fact that their precursors are stem/progenitor mesenchymal cells [88]. 

Overall, our data demonstrate that “pulsed hypoxia” activates the mesenchymal–epithelial transition in the cells with molecular phenotype of CAFs and tumorigenic potency. Such transformation is realized through the appearance of intermediate mesenchymal cells with a high aggressive phenotype. The resulting panel of cells undergoing MET at different stages can be a useful tool for studying the processes that regulate metastasis We tried to summarize all of the observed changes in Figure 8. Data demonstrate the increase/decrease in mRNA/protein level in intermediate mesenchymal cells compared to BrC4f cells.

## 4. Materials and Methods

### 4.1. Cell Culture 

MCF-7, BrC4 and BrC4e cells were cultured in Iscove’s Modified Dulbecco’s Medium (IMDM) (#I7633-10X1L, Sigma-Aldrich, St. Louis, MO, USA), MDA-MB-231 was cultured in Dulbecco’s Modified Eagle’s Medium (DMEM) (#D6046-1L, Sigma-Aldrich, St. Louis, MO, USA) and A431 was cultured in Dulbecco’s Modified Eagle Medium/Nutrient Mixture F-12 (DMEM/F12) (#42400028, Gibco™, New York, NY, USA). All cultures’ media were supplemented with 10% fetal bovine serum (FBS) (#A316040, Thermo Fisher, Waltham, MA, USA) and with 250 mg/mL amphotericin B and 100 U/mL penicillin/streptomycin (#15140122, Gibco™, Waltham, MA, USA). Cells were cultured at 37 °C with 5% CO_2_ unless other conditions are mentioned.

MCF-7 (#ACC 115, DSMZ, Braunschweig Germany), MDA-MB-231(#ACC 65, DSMZ, Braunschweig Germany) and A431 (ATCC, #CRL-1555) were purchased from the American Type Culture Collection (ATCC, USA), and BrC4 was obtained in the Laboratory of Biotechnology in the Institute of Chemical Biology and Fundamental Medicine SB RAS (ICBFM, Novosibirsk, Russia) [41]. All cell lines were detected free of mycoplasma contamination with RT-PCR. 

“Pulsed hypoxia” has been used to progressively transform BrC4f cells from mesenchymal to epithelial phenotypes. This method is based on the replacement of the cultivating condition from normoxia to hypoxia with the addition of the conditioned medium. The oxygen content was analyzed using a quadrupole mass spectrometer by the ratio of oxygen to nitrogen (Agilent 6490, Agilent Technologies, Santa Clara, CA USA). The rating of oxygen to nitrogen was 0.275 ± 0.005 (in control) and 0.256 ± 0.005 (one round of PH). Cells after the first and second rounds of “pulsed hypoxia” were named BrC4f_Hyp1 and BrC4f_Hyp2, respectively. The transformed cells BrC4e were cultured in standard cultivation conditions [41].

### 4.2. Total RNA Isolation and Reverse Transcriptase–Quantitative Polymerase Chain Reaction (RT-qPCR)

Cells were homogenized in the Lira reagent (Biolabmix, Novosibirsk, Russia), and RNAs were extracted following the protocol provided by the manufacturer. cDNAs were synthesized from the total RNAs using the M-MuLV–RH First Strand cDNA Synthesis Kit (Biolabmix, Novosibirsk, Russia). qPCR was performed on Bio-Rad CFX96 (Bio-Rad, Hercules, CA, USA). Amplification parameters were as follows: 95 °C for 5 min followed by 35 cycles of 95 °C for 10 s, 60 °C for 10 s, 72 °C for 10 s and an additional cycle for the melting curve. The mRNA level of *HPRT* was detected as a loading control. Significance in changes of gene expression was calculated based on experiments in triplicate using a two-tailed Student’s *t*-test. Results were presented as histograms with relative units of transcription levels calculated in CFX Maestro Software (Bio-Rad, Hercules, CA, USA). Primers for RT-qPCR are listed below in Table 1. The forward and reverse primers were synthesized in ICBFM SB RAS, Novosibirsk, Russia. The levels of mRNAs are represented as relative values normalized to the level of HPRT mRNA. Mean values (± standard deviation) from three independent experiments are shown.

### 4.3. Western Blot

Cells were lysed in the Tris-HCl (pH6.6) buffer containing 1% of SDS and a complete protease inhibitor cocktail (#11836145001, Roche Diagnostics GmbH, Mannheim, Germany). The protein concentration was measured using the Pierce™ Modified Lowry Protein Assay Kit (#23240, Thermo Fischer, Rockford, IL, USA). Samples (30 μg) were separated by 10% SDS-PAGE and transferred to a Trans-Blot nitrocellulose membrane (Bio-Rad Laboratories, Hercules, CA, USA) by a wet blotting procedure (100 V, 500 mA, 90 min, 15 °C) using “Mighty Small Transphor” (GE healthcare Bio-Science AB, Helsinki, Finland). Immunodetection was performed using the iBind system (Life Technologies), iBind Cards (Invitrogen, Thermo Fisher Scientific, Waltham, MA, USA) and antibodies: primary antibodies against tubulin (#T8328, Sigma-Aldrich, Burlington, MA, USA), vimentin (#ab8069, Abcam, Cambridge, UK), GAPDH (#39-8600, Thermo Fischer, Waltham, MA, USA), SNAIL + SLUG (#ab180714, Abcam, Cambridge, UK), HIF1α (#A16873, ABclonal, Woburn, MA, USA), HIF1β/ARNT (#A19532, ABclonal, Woburn, MA, USA) and HRP-conjugated secondary anti-rabbit (#31460, Thermo Fischer, Waltham, MA, USA) and anti-mouse (#31430, Thermo Fischer, Waltham, MA, USA) antibodies. The signals were developed using a Novex ECL HRP chemiluminescent substrate reagent kit (#WP20005, Invitrogen, Waltham, MA, USA) and detected with GE Amersham Imager 600 (GE, Marlborough, MA, USA). Densitometric analysis of the western blot data was performed using the image analysis software GelAnalyser version 2010a.

### 4.4. Flow Cytometery

Cells growing in T25 flasks (TPP, Trasadingen, Switzerland) were collected, fixed in 10% neutral buffered formalin (BioVitrum, Saint-Petersburg, Russia) and incubated with labeled mouse anti-human antibodies N-cadherin (#2354025, Sony Biotechnology Inc. San Jose, CA, USA), E-cadherin (#2220530, Sony Biotechnology Inc. San Jose, CA, USA), Ep-CAM (#2221070, Sony Biotechnology Inc. San Jose, CA, USA), CD44 (#560890, BD Pharmigen), CD90 (#ab134360, Abcam, Cambridge, UK), CD34/CD45 (#34107, BD Pharmigen, San Jose, CA, USA), CD31 (#8107781, BD Pharmigen, San Jose, CA, USA), CD73 (#344006, BioLegend, San Diego, CA, USA) and CD105 (#323204, Bio-Legend, San Diego, CA, USA) in PBS supplemented with 10% normal goat serum for 30 min in ice. All analyses were performed using a FACSCantoII flow cytometer (BD Biosciences, Franklin Lakes, NJ, USA), and the data were analyzed by FACSDiva Software Version 6.1.3. (BD Biosciences). Cells were initially gated based on forward versus side scatter to exclude small debris, and ten thousand events from this population were collected. Control cells were treated with appropriate isotype PE-conjugated IgG (BD Biosciences).

### 4.5. Aldefluor Assay

ALDH activity was determined using the activated AldefluorTM reagent, a fluorescent non-toxic substrate for ALDH1 able to freely diffuse into intact and viable cells (Stem Cells Technologies, Grenoble, France). A 1 × 10^6^ portion of cells was suspended in 1 mL of the Aldefluor assay buffer containing the ALDH1 substrate (Bodipy-Aminoacetaldehyde) and incubated for 45 min at 37  °C. As a reference control, the cells were suspended in the buffer containing the substrate in the presence of diethylaminobenzaldehyde (DEAB; 20 μM for the rest of the experiments), a specific ALDH1 enzyme inhibitor. All analyses were performed using FACSCantoII flow cytometer (BD Biosciences, Franklin Lakes, NJ, USA), and the data were analyzed by FACSDiva Software Version 6.1.3. (BD Biosciences). Cells were initially gated based on forward scatter versus side scatter to exclude small debris, and ten thousand events from this population were collected.

### 4.6. Cathepsin D Activity

MDA-MB-231, MCF-7, BrC4e and BrC4f cells were seeded in a 6-well plate and were grown to 80–90% confluence. BrC4f cells proceeded to undergo one or two rounds of hypoxia to obtain Hyp1 and Hyp2 cells before the cathepsin D activity test. Cells were harvested with the TrypLE™ Express enzyme (Gibco, Gaithersburg, MD, USA) for 5 min at 37 °C, and sample preparation was carried out according to the manufacturer’s instructions. The resulting mixtures were analyzed using a fluorimeter (Cary Eclipse, Varian, Australia) at Ex/Em = 328/400 nm using 5 nm excitation and emission slits. Cathepsin D activity in the cell lysates was assayed using a Cathepsin D activity assay kit (Abcam, Cambridge, UK, #ab65302). The kit is a fluorescent-based assay that utilizes the preferred Cathepsin D substrate sequence GKPILFFRLK(Dnp)-D-R-NH2) labeled with methyl coumaryl amide (MCA). The total protein level in cell lysates was also measured using a Modified Lowry Protein Assay Kit (Thermo Fisher, Waltham, MA, USA, #23240), and relative level of fluorescence (RFU) obtained in Cathepsin D activity assay was recalculated according to protein level in cell lysates.

### 4.7. Immunocytochemistry

Cells (1  ×  10^4^) growing in four-well culture slides (BD Falcon, Bedford, MA) were washed with PBS and 10% neutral-buffered formalin. To block nonspecific antibody binding, cells were incubated in 1% BSA (Sigma-Aldrich, Burlington, MA, USA) and 0.3 M glycine in PBST buffer (PBS with 0.1% Tween 20) for 30 min at RT. Next, cells were incubated with the VIMENTIN (#ab8069, Abcam, Cambridge, UK) and E-CADHERIN (#2220530, Sony Biotechnology Inc., San Jose, CA, USA), S100A4 (#MA5-31333, Invitrogen, Waltham, MA, USA) and αSMA (#ab5694, Abcam, Cambridge, UK) antibodies for 60 min at RT. For visualization, FITC-conjugated (#ab97050, Abcam, Cambridge, UK) or Alexa Fluor 555-conjugated (#A32727, Invitrogen, Waltham, MA, USA) secondary antibodies were used for 1 h at RT. Stained cells were visualized using an Axioscop 2 PLUS fluorescence microscope (Carl Zeiss, GmbH, Jena, Germany) or inverted microscope (Eclipse Ti, Nikon, Tokio, Japan).

### 4.8. Xenograft Assay 

Female SCID hairless outbred (SHO-PrkdcscidHrhr) mice aged 6–8 weeks were obtained from the SPF vivarium of the Institute of Cytology and Genetics of the Siberian Branch of the Russian Academy of Science (SB RAS) (Novosibirsk, Russia). Mice were housed in individually ventilated cages (Animal Care Systems, Colorado, USA) in groups of 1–2 animals per cage with ad libitum food (ssniff, Soest, Germany) and water. Mice were kept in the same room within a specific pathogen-free animal facility with a regular 14/10 h light/dark cycle (lights on at 02:00 h) at a constant room temperature of 22  ±  2 °C and relative humidity of approximately 45  ±  15%.

All animal experiments were carried out in compliance with the protocols and recommendations for the proper use and care of laboratory animals (EEC Directive 86/609/EEC). The study protocol was approved by the Committee on the Ethics of Animal Experiments of the Administration of SB RAS (Permit #40 from 4 April 2018). 

A suspension of BrC4f and BrC4f_Hyp2 (2.5* ×  10^7^ cells/mL) in PBS mixed with Matrigel (BD Bioscience) at a ratio of 1:1 and 0.1 mL of suspension were injected subcutaneously into the dorsal flank of mouse. Tumor volume was measured twice a week. The significance of difference in tumor volumes between two groups was calculated using two-way ANOVA tests. After extraction, tumor tissues were fixed in 10% neutral-buffered formalin, processed and embedded into paraffin blocks.

### 4.9. Hematoxylin and Eosin Staining 

Paraffin blocks were sectioned to 4 µm, deparaffinized and dehydrated. Afterwards the sections were stained with hematoxylin and eosin using convectional protocol. The slides were visualized with Axioscop 2 PLUS (Carl Zeiss, GmbH, Jena, Germany) at the Multiple-access Center for Microscopy of Biological Subjects (Institute of Cytology and Genetics, Novosibirsk, Russia).

### 4.10. Tissue Samples Immunostaining 

The slides were deparaffinized in xylene, rehydrated in grade ethanol and washed with 0.1% tween in PBS (PBS-T). For antigen retrieving, the slides were heated to 121 °C in an autoclave for 20 min in a citrate buffer (10 mmol/L, pH 6.0). The endogenous peroxidase was blocked by immersing the slides in 3% H_2_O_2_ for 10 min. Subsequently, the slides were incubated with anti-C-MYC primary antibodies (MAB36961, R&D systems) at 4 °C overnight. After washing, each slide was incubated with HRP-conjugated anti-rabbit secondary antibodies (#31460, Thermo Fischer Waltham, MA, USA). Afterwards, the slices were incubated with the 3,39-diaminobenzidine tetrahydrochloride (DAB) reagent for 10 min. Then, hematoxylin re-staining was performed, followed by dehydration, clearing and mounting. For the negative control, slides were incubated with PBS -T instead of primary antibodies. The slides were visualized with Axioscop 2 PLUS (Carl Zeiss, GmbH, Jena, Germany) at the Multiple-access Center for Microscopy of Biological Subjects (Institute of Cytology and Genetics, Novosibirsk, Russia).

### 4.11. Statistics

Significance was determined using a two-tailed Student’s *t*-test. A *p* value of less than 0.05 was considered significant. All the error bars represent the standard error of the mean. For normal data distribution, the significance was determined using a two-tailed Student’s *t* test. For non-normal data distribution, the Mann–Whitney u-test was used.

## 5. Conclusions

We have demonstrated that pulsed hypoxia induced transformation of fibroblast-shaped cells with molecular phenotype of CAFs to semi-epithelial cells. Such conditions activate the malignant progression in the cells with tumorigenic potency. This malignant progression is realized through the appearance of intermediate mesenchymal cells. Western blot analysis, fluorescent microscopy and flow cytometry of transformed cells demonstrated the decrease in mesenchymal markers and an increase in the epithelial markers in the cells with intermediate phenotypes. Although protooncogene c-MYC expression increased in the final cells with an epithelial phenotype, cells with an intermediate phenotype were the most aggressive. It is possible that such cells become tumor-initiating cells during metastasis in vivo. We suggest that regulation of the oxygen level for stromal areas in a tumor can be a strategy for the development of new antitumor approaches. Our cell models would be useful tools for testing such approaches. 

## Figures and Tables

**Figure 1 ijms-24-02494-f001:**
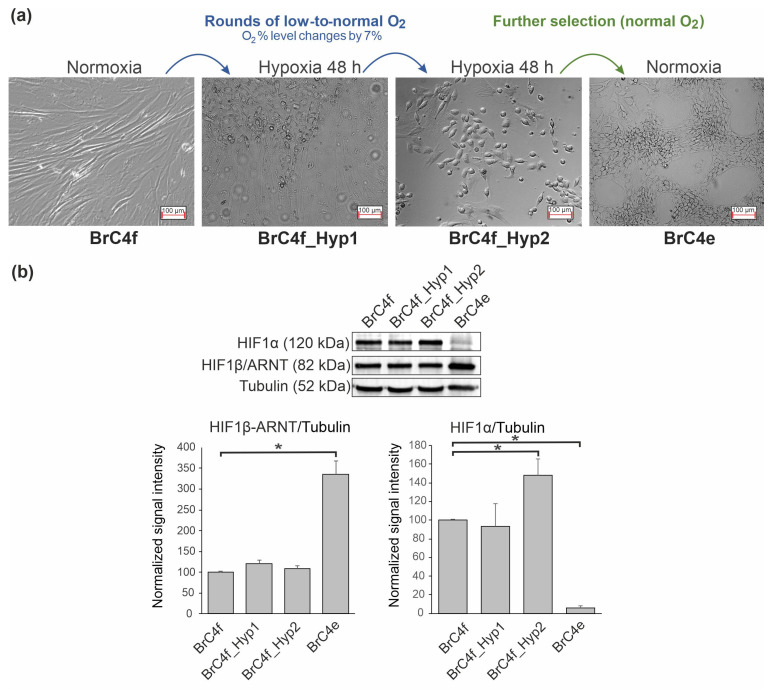
(**a**) Inline transformation of BrC4f cells under rounds of pulsed hypoxia. Cell morphology was monitored by phase-contrast microscopy. (**b**) Western blots analysis of HIF-1β/ARNT and HIF-1α in cells. Beta-tubulin was used as a loading control. The differences were significant with * *p* < 0.05 between two groups.

**Figure 2 ijms-24-02494-f002:**
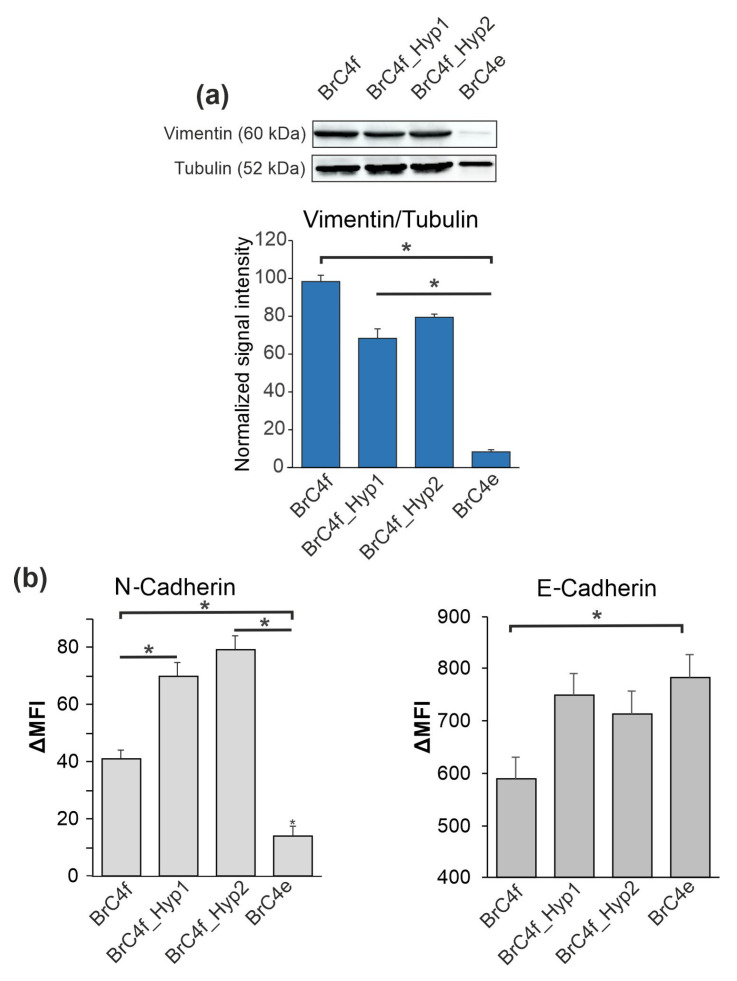
Changes in epithelial and mesenchymal molecular markers in cell cultures after rounds of hypoxia. BrC4f, BrC4f_Hyp1, BrC4f_Hyp2 and BrC4e cells were analyzed. (**a**) Western blot analysis of vimentin in cell lysates; tubulin was used as loading control. (**b**) Flow cytometry analysis of the surface-located N-cad and E-cad. Data are presented as Mean Fluorescence Intensity (MFI) ± SD. The differences were significant with * *p* < 0.05 between two groups.

**Figure 3 ijms-24-02494-f003:**
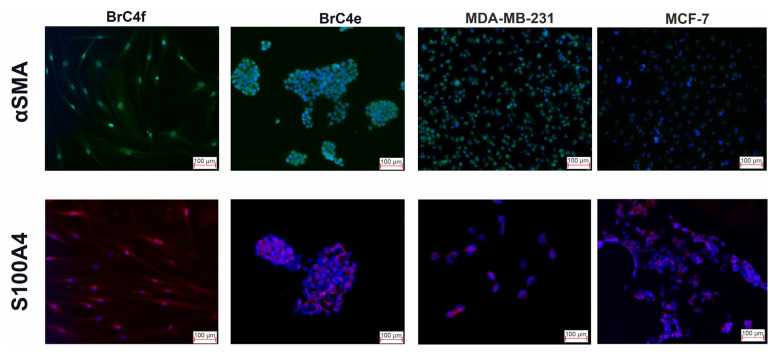
Immunofluorescence staining for cytoskeletal markers. FITC-conjugated anti-αSMA (green signal) and Alexa Fluor 555-conjugated anti- S100A4 (red signal) antibodies were used.

**Figure 4 ijms-24-02494-f004:**
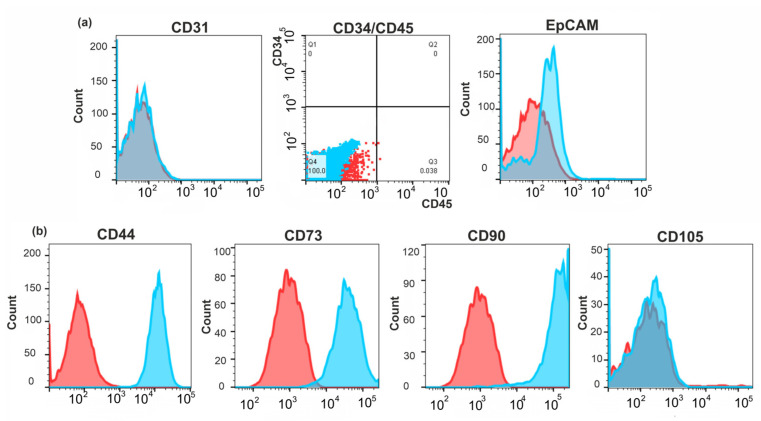
Markers used in the identification of profiling CAFs by flow cytometry. (**a**) Negative control CD34—transmembrane phosphoglycoprotein; CD45—protein tyrosine phosphatase, receptor type, CD31—platelet endothelial cell adhesion molecule and EpCAM–epithelial cell adhesion molecule; (**b**) CD90—membrane GPI-anchored protein with one Ig V-type superfamily domain; CD73—ecto-5-nucleotidase; CD105—surface endoglin; C. Red—negative control, blue—experiment.

**Figure 5 ijms-24-02494-f005:**
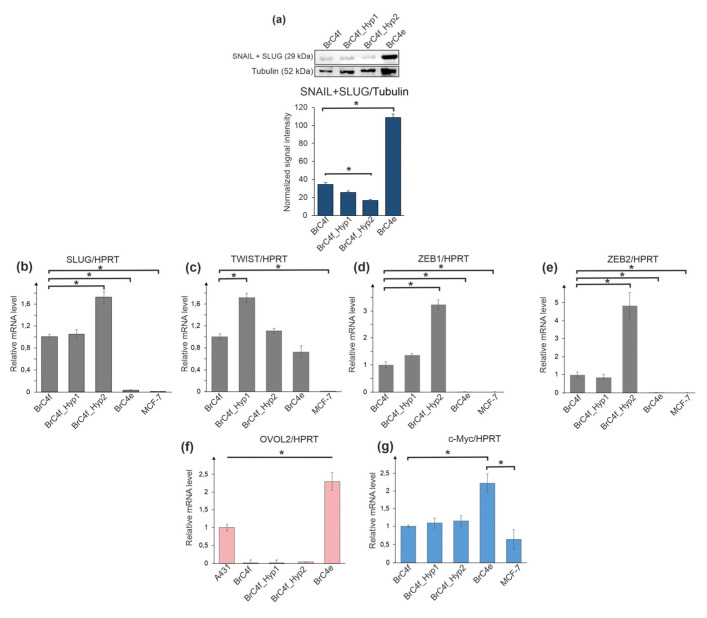
The analysis of markers of EMT and MET in model cells. (**a**) Protein levels of SNAIL/SLUG in cell culture transformation of fibroblasts. Quantitation of SNAIL/SLUG normalized by total beta-tubulin. * *p* < 0.05. Alteration of mRNA expression level genes associated with epithelial–mesenchymal transition and mesenchymal–epithelial transition. (**b**) SLUG; (**c**) TWIST1; (**d**) ZEB1; (**e**) ZEB2; (**f**) OVOL2; (**g**) c-MYC. The expression of specific mRNAs was normalized to the expression level of Hypoxanthine Phosphoribosyl transferase (HPRT) mRNA. Statistical analysis included the results of two independent experiments (mean  ±  SD). * The difference between the experimental group and the control (BrC4f) group were statistically significant at *p* < 0.05.

**Figure 6 ijms-24-02494-f006:**
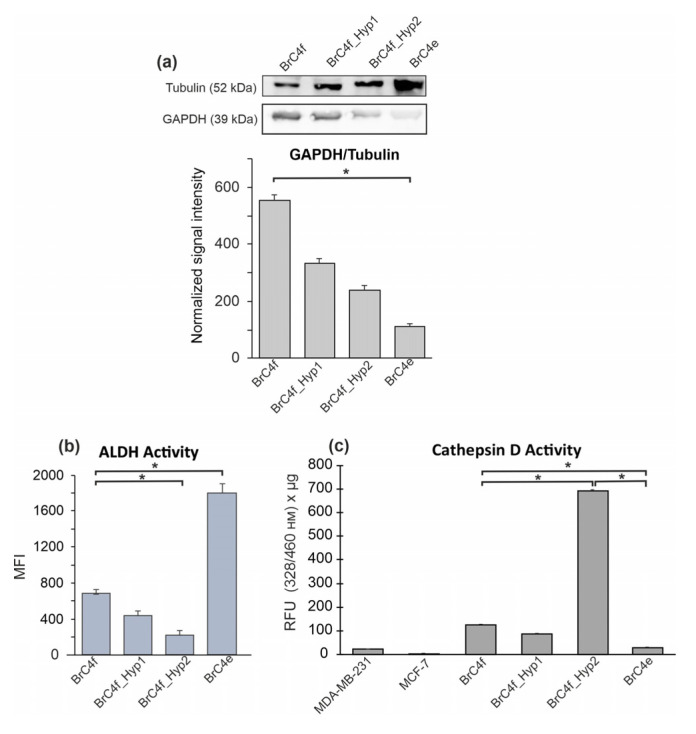
Analysis of metabolic enzymes in established cell lines. (**a**) Western blot analysis of GAPDH level. (**b**) Alterations of ALDH activity in the cells. ALDEFLUOR™ kit assay measurements of ALDH activity by flow cytometry. ALDH converts the ALDH substrate, BAAA(BODIPY-aminoacetaldehyde), into the fluorescent product BAA-(BODIPY-aminoacetate), which is retained inside viable cells and detected through use of the green fluorescence channel (emission wavelength of 585–595 nm). Flow cytometry data are presented as mean fluorescent intensity (MFI) of the cells ± SD. (**c**) CatD activity in the cells. MDA-MB-231 cells and MCF7 cells were used as internal control. The data are presented as mean value ± SD of cellular fluorescence in relative fluorescent units (RFU) per microgram of total protein amount in the sample. Data are of three independent experiments. The differences between two groups were significant (* *p* < 0.05).

**Figure 7 ijms-24-02494-f007:**
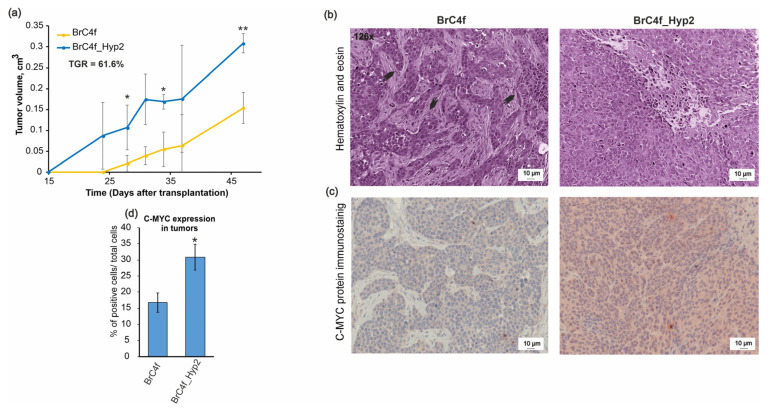
The tumorigenic properties of established cancer cells in vivo. (**a**) Comparison of tumor growth formed by fibroblasts transplanted into SCID mice. Tumor growth was monitored twice a week and plotted as tumor volume against time. Data are presented as mean ± SD (*n* = 6 per group); statistical analysis was performed using one sample Student *t*-test. (*) *p* < 0.05, (**) *p* < 0.01. (**b**) Histological analysis of tumors forming by BrC4f and BrC4f_Hyp2. H&E staining. Fibrotic focus indicated by arrow. (**c**) Quantification of the c-MYC oncoprotein in BrC4f and BrC4f_Hyp2 tumors. (**d**) c-MYC detection in BrC4f and BrC4f_Hyp2 tumor samples.

**Figure 8 ijms-24-02494-f008:**
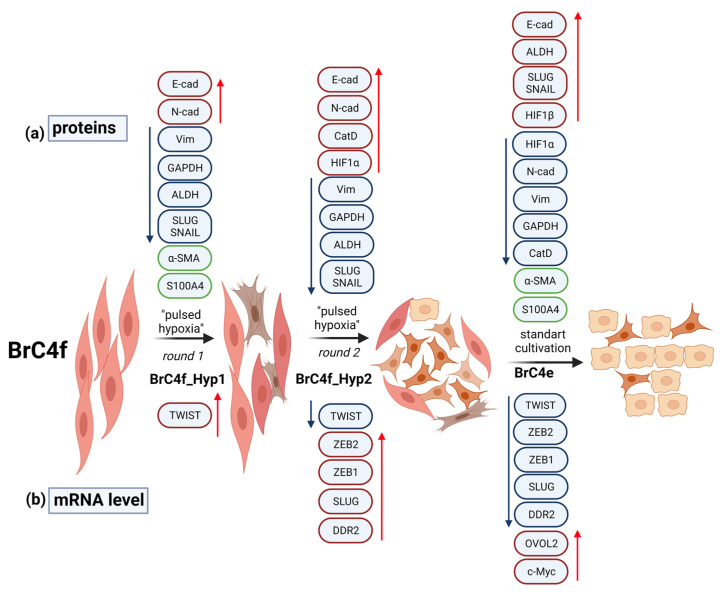
Resulting scheme of undergoing MET at different stages of breast CAFs by “pulsed hypoxia”. Changes in MET markers of (**a**) proteins and (**b**) genes. Red arrows indicate the increase, and blue arrows indicate the decrease, in markers compared to BrC4f cells (control). Green boxes indicate stable state of marker.

**Table 1 ijms-24-02494-t001:** Primers for RT-qPCR.

Target Gene	Sequence 5‘→3‘	Primer Length
*TWIST1*	F: GGCATCACTATGGACTTTCTCTATT R: GGCCAGTTTGATCCCAGTATT	25 21
*SLUG*	F: TGGTTGCTTCAAGGACACAT R: GCAAATGCTCTGTTGCAGTG	20 20
*ZEB2*	F: CGATCCAGACCGCAATTAAC R: TGCTGACTGCATGACCATC	20 19
*ZEB1*	F: AACTGCTGGGAGGATGACAC R: TCCTGCTTCATCTGCCTGA	20 19
*OVOL1*	F: ACGATGCCCATCCACTACCTG R: TTTCTGAGGTGCTGGTCATCATTC	21 24
*OVOL2*	F: GGCAAGGGCTTCAACGACA R: CTTCAGGTGGGACTCCAGAGA	19 21
*C-MYC*	F: CTTCTCTCCGTCCTCGGATTCT R: GAAGGTGATCCAGACTCTGACCTT	22 24
*HPRT*	F: CATCAAAGCACTGAATAGAAAT R: TATCTTCCACAATCAAGACATT	22 22

## Data Availability

Not applicable.

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
