# Peer review of "“Pulsed Hypoxia” Gradually Reprograms Breast Cancer Fibroblasts into Pro-Tumorigenic Cells via Mesenchymal–Epithelial Transition"

_ijms, 2023, doi:10.3390/ijms24032494_

Round 1
Reviewer 1 Report
Dear colleagues!
After review of the manuscript by Nushtaeva et al. I have the following comments as a peer reviewer assigned by the Office:
1) Generally, it is a well-designed study imposing a valid hypothesis and topic of mechanisms of tumor progression and growth implying the role of hypoxia. The paper is well-written yet has minor spelling and typo errors which I indicate below.
2) One crucial point the might be questioned by the Reader is level of HIF response to indicted levels of hypoxia. Is there and data on change of HIFs in BRC4F used that might be obtained by WB or other means to validate the response especially taking into consideration that "pulse" approach was used and the 1st pulse may pre-condition the cells prior to the 2nd one?
3) In Fig 1 I suggest to provide O2% level to clearly indicate the hypoxia indicted on cultured cells
4) Fig. 4 I suggest to indicate aSMA and S100A4 at the left side of panel in vertical orientation to clearly discern between lines and columns. Current indication is a bit confusing
5) In Fig. 4 CD105 might be place in line with other MSC positive marker s(CD73 and 105)
6) Typically in WB loading control (tubulin) is placed below the target protein (Fig.5 and 6)
7) the Discussion section would definitely benefit from being shortened and more focused on findings of the study.
8) In line 175 "Phenotype cell BrC4f" looks misspelled
9) In line 160-161 I suggest to avoid capital lettering in SMA an S100 deciphering in text
10) In line 103 propOgated is definitely misspelled (correct is propagAted)
Regards, Reviewer
Author Response
Point 1: Generally, it is a well-designed study imposing a valid hypothesis and topic of mechanisms of tumor progression and growth implying the role of hypoxia. The paper is well-written yet has minor spelling and typo errors which I indicate below.
Response 1: Thank you for your kind consideration of our work. In the current version of the manuscript we corrected minor spelling and typo errors.
Point 2: One crucial point the might be questioned by the Reader is level of HIF response to indicted levels of hypoxia. Is there and data on change of HIFs in BRC4F used that might be obtained by WB or other means to validate the response especially taking into consideration that "pulse" approach was used and the 1st pulse may pre-condition the cells prior to the 2nd one?
Response 2: In the current version of the manuscript we added data of HIFs level in the cells (WB) (See Fig. 1 and See Lines 146-152) and discussed this results (See Lines 374-385).
Point 3: In Fig 1 I suggest to provide O2% level to clearly indicate the hypoxia indicted on cultured cells.
Response 3: In the current version of the manuscript we added experimental data of the oxygen content in medium (See Lines 131-132 and Fig. 1). In the Materials and Methods we described how the oxygen content was analyzed (See Lines 507-510).
Point 4: Fig. 4 I suggest to indicate aSMA and S100A4 at the left side of panel in vertical orientation to clearly discern between lines and columns. Current indication is a bit confusing.
Response 4: In the current version of the manuscript we indicated aSMA and S100A4 at the left side of panel in vertical orientation (See Line 189) in Fig. 3.
Point 5: In Fig. 4 CD105 might be place in line with other MSC positive marker s(CD73 and 105).
Response 5: In the current version of the manuscript we moved CD105 in line with other MSC positive markers (See Line 202).
Point 6:Typically in WB loading control (tubulin) is placed below the target protein (Fig.5 and 6).
Response 6: In the current version of the manuscript we moved WB loading control (tubulin) below the target protein (Fig.5 and 6) (See Lines 231 and 263).
Point 7: the Discussion section would definitely benefit from being shortened and more focused on findings of the study.
Response 7: In the current version of the manuscript we shortened the Discussion section and focused on findings of the study. However, according to the recommendation of reviewer 2, we added an additional paragraph of discussion (See Lines 462-481).
Point 8: In line 175 "Phenotype cell BrC4f" looks misspelled.
Response 8: The misspelling: line 198 is now corrected in the revised manuscript.
Point 9: In line 163 I suggest to avoid capital lettering in SMA an S100 deciphering in text.
Response 9: In the current version of the manuscript, the capital letters in the SMA and S100 transcript have been corrected to lower case (See Lines 181-182).
Point 10: In line 103 propOgated is definitely misspelled (correct is propagAted)
Response 10: The misspelling: line 114 is now corrected in the revised manuscript.

Reviewer 2 Report
The authors studied that pulsed hypoxia culturing transformed the cancer-associated fibroblasts and become epithelial-like tumor cells. Overall, the findings are interesting with some concerns as follows:
1. I am confusing the tested cell line (BRC4f cell) isolated from a single cell clone? Is BRC4f cell a breast cancer cell line (https://doi.org/10.1186/s12935-019-0766-5) or a cancer-associated fibroblast. Expressions of CD90, CD73, vimentin and N-cadherin also reported in MDA-MB-231 cell or other breast cancer cell. The reviewer did not understand why authors determinate BRC4f is a fibroblast cell when BRC4f cells showed the in vivo tumor formation ability?
2. The authors compared the in vivo tumor formation of BrC4f and BrC4f_Hyp2 cells, but not BrC4e cells. Is BrC4e cell able to tumor formation?
3. The connect between EMT/MET, tumor growth rate and c-Myc expression is missing.
Author Response
Point 1: 1. I am confusing the tested cell line (BRC4f cell) isolated from a single cell clone? Is BRC4f cell a breast cancer cell line (https://doi.org/10.1186/s12935-019-0766-5) or a cancer-associated fibroblast. Expressions of CD90, CD73, vimentin and N-cadherin also reported in MDA-MB-231 cell or other breast cancer cell. The reviewer did not understand why authors determinate BRC4f is a fibroblast cell when BRC4f cells showed the in vivo tumor formation ability?
Response 1: We thank the reviewer for the positive feedback on our manuscript. BRC4f breast cancer cells line were isolated from tumor tissue (https://doi.org/10.1186/s12935-019-0766-5). The fibroblastoid-like BrC4f cells has spindle-like morphology. In accordance with the reviewer's comment, we have become more cautious about stating that these cells are CAFs and do not exclude that they have similarities with the tumorigenic mesenchymal cells MDA-MB-231 (See Lines 355; 486-489).
The analysis of two additional molecular markers were added to the manuscript - endothelial (CD31) and epithelial (EpCAM) markers (See Figure 4). Moreover, In the current version of the manuscript, we have added information on the determination of CAFs in the introduction (See Lines 65-70; 67-69).
In the current version of the manuscript, we have added to the discussion about the expression of mesenchial markers that have also been reported in MDA-MB-231 cells or other breast cancer cells (See Lines 468-481).
Several studies have shown that there are rare forms of stromal breast cancer (DOI: 10.7759/cureus.5143 DOI: 10.3233/BD-201012) We can't say with certainty that the BrC4f cells as a CAFs or other breast cancer cell with mesenchymal-like phenotype with high stem cells potency. To confirm our hypothesis that BrC4f_Hyp2 acquired a more aggressive phenotype than parental cells, in vivo tumorigenicity studies of BrC4f_Hyp2 and BrC4f were performed.
Point 2: The authors compared the in vivo tumor formation of BrC4f and BrC4f_Hyp2 cells, but not BrC4e cells. Is BrC4e cell able to tumor formation?
Response 2: We previously showed that BrC4e cells do not contribute to tumor formation (https://doi.org/10.1186/s12935-019-0766-5) and repeated this information in the current version (See Lines 280-281).
Point 3: The connect between EMT/MET, tumor growth rate and c-Myc expression is missing.
Response 3: In the current version of the manuscript we added connection between EMT/MET, tumor growth rate and c-Myc expression (See Lines 462-467).

Round 2
Reviewer 2 Report
My major concern is the previous study did not rule out CAF and tumor cells existed together in your culture. When culture hypoxia, CAF become great majority cell.
Author Response
Point 1: My major concern is the previous study did not rule out CAF and tumor cells existed together in your culture. When culture hypoxia, CAF become great majority cell.
Response 1: Thank you for the suggestion. We assume that BrC4f cells could be changes by EMT in patient during cancer progression. So, we can’t discard hypothesis that BrC4f are tumor cells. However, based on our results we can only say that BrC4f demonstrate the whole set of CAF markers and produce tumor in vivo.